# An Investigation into the Influence of Different Types of Nesting Materials upon the Welfare of Captive Chimpanzees (*Pan troglodytes*)

**DOI:** 10.3390/ani11061835

**Published:** 2021-06-20

**Authors:** Naomi Anderson, Bala Amarasekaran, David Riba

**Affiliations:** 1Fundació UdG: Innovació i Formació, Universitat de Girona, 17003 Girona, Spain; 2Tacugama Chimpanzee Sanctuary, Freetown 47235, Sierra Leone; Bala@tacugama.com; 3Facultat de Lletres, Universitat de Girona, 17003 Girona, Spain; david.ribacano@udg.edu

**Keywords:** bedding, nesting materials, environmental enrichment, welfare, captive chimpanzees

## Abstract

**Simple Summary:**

Presently, it is vital that captive chimpanzee populations are kept physically and mentally healthy, considering the number of individuals remaining in the wild continues to decrease. As such, we carried out this study to further understand how different types of nesting materials impact upon the welfare of captive chimpanzees living at Tacugama Chimpanzee Sanctuary (Sierra Leone). These nesting materials, including leaves and branches, long grass, cotton sheets, and shredded newspaper, served as a form of environmental enrichment—a strategy that is often adopted to enhance animals’ welfare. We found that the chimpanzees’ welfare was higher when nesting materials were present, compared to when they were absent, and that individuals favoured shredded newspaper. Our results reinforce the importance of supplying enrichments that are destructible, as these give individuals an increased sense of control, and simultaneously demonstrate that synthetic enrichment materials can be favoured over natural ones.

**Abstract:**

Ensuring the welfare of captive chimpanzees (*Pan troglodytes*) is crucial, in part because they can act as a conservation resource in the case of wild populations becoming extinct. One strategy often adopted to ensure animal welfare is environmental enrichment. In this study, we investigated the impact of different nesting materials (leaves and branches, long grass, cotton sheets, and shredded newspaper) upon the welfare of chimpanzees housed at Tacugama Chimpanzee Sanctuary (Sierra Leone). Data was collected on 20 subjects (aged 4 to 15 years) between November 2019 and March 2020. Welfare was based on individuals’ relative frequency of affiliative, abnormal and agonistic behaviours, as well as their social and behavioural competence. We found that individuals’ welfare was higher when presented with nesting materials (compared to the control and post-treatment conditions), particularly when given shredded newspaper, regardless of whether the material was presented separately or in conjunction with another. In addition, welfare was highest: during the morning (vs evening); in groups of older individuals (vs younger); with females engaging in less agonistic behaviours compared to males; and males displaying relatively higher behavioural competence. Our results support previous research that captive chimpanzees be supplied with destructible nesting materials, and demonstrate that synthetic enrichments can indeed have a more positive impact upon welfare than their natural alternatives.

## 1. Introduction

Chimpanzees (*Pan troglodytes*) have been listed as endangered by the IUCN for two decades [1]. As such, attention is directed towards conserving the 345,000–470,000 individuals estimated to be left in the wild [1]. Regrettably, in-situ conservation methods are struggling to protect this species due to hunting [2], logging and forest clearance [3], the development of road networks [4], amongst other reasons. The chimpanzees currently held in captivity are an important conservation resource in the event of wild populations becoming extinct [5]. Thus, ensuring captive chimpanzees’ welfare needs are met is essential, not only for their health and comfort while away from their natural habitat, but also to safeguard populations that can potentially be re-introduced into the wild.

In an attempt to enhance physical and/or psychological welfare [6], animals held in captivity are often supplied with environmental enrichment. The addition of novel stimuli to an individual’s captive environment aims to increase behavioural choices, whilst simultaneously decreasing boredom and inactivity [7]. Although some studies have found that natural—rather than synthetic—materials are often more effective at improving animal welfare [8], when effectiveness is assessed [9], environmental enrichment can include anything from items that increase foraging time to the active participation in cognitive based research [10,11,12]. Distinct species have different behavioural patterns, so it is imperative to supply an enrichment that draws out species-specific and appropriate behaviours [7].

The provision of nesting materials is one way of improving primate welfare [13], which can be viewed as a form of environmental enrichment. For instance, when rhesus macaques (*Macaca mulatta*) were presented with pine shavings, their rates of overgrooming and aggression reduced, whilst their time spent foraging increased [14]. Similarly, deep woodchip litter decreased aggressive behaviours and increased play in eight primate species, including black-capped capuchins (*Cebus apela*), stumptail macaques (*Macaca arctoides*) and common marmosets (*Callithrix jacchus*) [15]. Although captive facilities regularly provide bedding for their chimpanzees, few studies have assessed its impacts upon chimpanzee welfare. The limited published assessments include Brent (1992) [16] who found that supplying woodchip bedding to juvenile chimpanzees significantly reduced their abnormal behaviours, and Baker (1997) [17] who reported that the use of straw as a floor covering reduced rates of regurgitation and reingestion, while increasing play, in adult chimpanzees. According to Fruth and Hohmann (1996) [18], nest construction forms a fundamental part of chimpanzees’ behavioural repertoire. Therefore, it is essential we have a good understanding of the impacts that nesting materials have upon the behaviour and welfare of captive chimpanzees.

Although there are few studies assessing the impacts of nesting materials upon chimpanzee welfare, researchers have indeed investigated their preference towards said materials, including Videan (2006) [19] who found that 73 captive chimpanzees housed at the University of Texas, engaged in more bed-building behaviour when presented with hay, compared to when the same subjects were offered paper or browse. Researchers have additionally found that captive great apes often combine their preferred nesting materials with other less favoured ones to build a nest [20]—a finding that has led recent studies to recommend the proviso of a combination of bedding materials to captive chimpanzees, where at least one is known to be preferred [21].

Despite the existence of publications investigating the effects of nesting materials upon captive chimpanzee welfare, and others reporting individuals’ preference towards said materials, there is a dearth of studies investigating both topics simultaneously. As such, this study investigates which materials are favoured by captive chimpanzees, by assessing those that produce the most positive impact upon on the subjects’ welfare, whilst simultaneously providing a recent assessment of the effectiveness of nesting materials as a form of enrichment for captive chimpanzees.

By providing different types of nesting materials (leaves and branches, long grass, cotton sheets, and shredded newspaper) to a group of captive western chimpanzees (*Pan troglodytes verus*) at Tacugama Chimpanzee Sanctuary (TCS), Sierra Leone, the first aim of our study was to investigate whether the provision of nesting materials improved individuals’ welfare. We predicted that (1) individuals’ welfare would increase when bedding materials were present and decrease when they were absent. The second aim was to assess which material enhanced welfare the most. The materials tested were leaves and branches, long grass, cotton sheets and shredded newspaper. We predicted that (2) the chimpanzees’ welfare would be most enhanced when receiving leaves and branches, seeing as this is the material used by wild individuals who bend and break larger and smaller branches of preferred tree species to construct their nests [22,23]. The third aim was to evaluate if the chimpanzees’ welfare differed when supplied with two materials, instead of one: leaves and branches with long grass, and cotton sheets with shredded newspaper. We predicted that (3) individuals would present greater welfare when given one of the combinations of bedding materials, compared to when the same materials were presented separately [20,21].

In addition, we predicted that the impact of bedding upon chimpanzee welfare would vary according to time of day, sex, group, age and group × age. We expected (4) individuals’ welfare to improve most during the evening, since the materials would be fresh and new, and boredom towards them could develop by the morning. When presented with the nesting materials, we expected (5) females’ welfare to increase more than that of males’, considering that female chimpanzees make nests more frequently throughout the day and spend more time engaged in the nest building process [24,25]. We also expected (6) younger chimpanzees’ welfare to increase more than older chimpanzees’, because younger individuals have been reported to make (day) nests more frequently than adults, often as a form of play [19,26]. As such, we also anticipated that (7) different groups would achieve different levels of welfare, particularly we expected (8) groups composed of younger individuals to show a greater improvement in welfare than groups composed of older chimpanzees.

Documenting how chimpanzees respond and react to nesting materials when in captivity, considering both individual and group differences, will help caregivers and captive facilities further appreciate the welfare needs of their residents and can consequently ensure these are being met.

## 2. Materials and Methods

All procedures in this experiment were carried out in accordance with the Tacugama Chimpanzee Sanctuary Ethics Committee guidelines (the project was given the approved research number of CS005).

This research was conducted with the approval of the resident vet of TCS, which in turn adheres to all animal protection and ethical regulations established by the Pan African Sanctuary Alliance (PASA). Please note that TCS does not always have the means to provide daily bedding to the chimpanzees due to difficulty in sourcing sustainable nesting materials (e.g., not deforesting the national park around the sanctuary to collect leaves and branches), lack of funds to purchase alternative materials, in addition to a lack of time and staff for extra cleaning the daily bedding would generate. As such, days on which the subjects did not receive bedding was not to purposefully reduce the standards of their housing conditions. Instead, the provision of bedding was seen as a bonus because during the dry season (when individuals spend more time outdoors) it is infrequently supplied.

### 2.1. Study Site

This study took place at TCS, located in the Western Area Peninsula National Park (WAPNP), in Freetown, Sierra Leone. TCS is a rescue and rehabilitation centre for western chimpanzees (*Pan troglodytes verus*), most of which were affected by the illegal bushmeat or pet trade. At the time of data collection TCS housed 99 chimpanzees, split into 10 social groups with indoor and outdoor enclosures. Individuals are indoors overnight where they have hammocks and platforms, and spend the daytime in the forested outdoors with access to diverse trees, ponds, wooden climbing structures and ropes.

### 2.2. Study Sample

We observed four out of the ten groups where access during morning and evening was possible, resulting in a total of 20 study subjects. Their age ranged from 4 to 15 years, thus including infants and juveniles/young adults, with 10 females and 10 males (Table 1).

### 2.3. Study Materials

The bedding materials we provided were: leaves and branches from diverse tree species within the WAPNP (Treatment 1), fresh long grass collected from natural grasslands (Treatment 2), small segments of cotton sheets (Treatment 3) and shredded newspaper (Treatment 4), chosen due to their low cost and ample availability in Sierra Leone. During the final two weeks of data collection, each group was presented with leaves and branches with long grass (Treatment 5), then cotton sheets with shredded newspaper (Treatment 6) (Table 2). The materials were added into the indoor enclosures whilst individuals were outdoors, just before their return inside at the end of the day and were scattered around the enclosure so each individual would have access to the material upon entry. The materials were fresh and new at each introduction and were cleared the following morning after the individuals were released back outside. As such, before each evening session of data collection, the chimpanzees were supplied with fresh nesting materials.

### 2.4. Data Collection

To assess the effect of bedding upon the chimpanzees’ welfare, we followed the repeated measures design [16,27], whereby observational data were collected under three conditions: before bedding (control condition), with bedding (during treatment condition) and after bedding (post treatment condition). Each condition lasted two days, with data collected in the evening, upon the individuals’ entry indoors, and the following morning, before they were released to their outdoor enclosures. To obtain the behavioural data on each of the six bedding treatments, we observed individuals over the course of six days per treatment. Although we aimed to collect data over six consecutive days, there were a few instances where a day of data collection was missed between stages, simply due the study taking place in a working chimpanzee sanctuary where many unpredictable occurrences arise.

The behaviours we recorded are listed in the ethogram (Appendix A).

### 2.5. Sampling Method

We used a multifocal scan sampling approach to collect the behavioural data. Each period of observation was split into sessions of 20 min (i.e., 20 min scans), during which each group member’s behaviour was recorded every 30 s. We aimed to collect 10 control scans (5 evening and 5 morning), 10 treatment scans (5 evening and 5 morning) and 10 post-treatment scans (5 evening and 5 morning) per individual. On a few occasions only eight scans were collected; however, due to the posterior data treatment and calculations performed, we do not believe this will have significantly impacted the results.

### 2.6. Schedule

The behavioural data was collected between November 2019 and March 2020. Individuals were sampled during 1 h 40 min in the evening after 3:30 p.m., and during 1 h 40 min in the morning before 10:30 a.m.

### 2.7. Measures of Welfare

We assessed individuals’ welfare by monitoring their behavioural changes with two distinct calculations: 1. relative frequencies and 2. welfare indices.

Firstly, we investigated individuals’ welfare by analysing the relative frequency of three behavioural categories: affiliative (including all sexual interactions), abnormal and agonistic. Welfare was believed to be higher when there were more affiliative behaviours [15] and less abnormal and agonistic behaviours [14,17]

Moreover, we explored individuals’ welfare by calculating two indices: their social competence index (SCI) and their behavioural competence index (BCI). Social competence refers to how well individuals handle interactions with their conspecifics [28]. Behavioural competence looks at the balance between individuals’ undesirable actions and their positive species-typical behavioural patterns [29].

As suggested by Llorente et al. (2015) [29], the formula used to calculate the SCI was
(1)$$\frac{\left(\mathrm{social}\;\mathrm{behaviours}\;-\;\mathrm{individual}\;\mathrm{behaviours}\right)}{\left(\mathrm{social}\;\mathrm{behaviours}\;+\;\mathrm{individual}\;\mathrm{behaviours}\right)}$$
and the formula used to calculate the BCI was
(2)$$\frac{\left(\mathrm{positive}\;\mathrm{individual}\;\mathrm{behaviours}\;-\;\mathrm{negative}\;\mathrm{individual}\;\mathrm{behaviours}\right)}{\left(\mathrm{positive}\;\mathrm{individual}\;\mathrm{behaviours}\;+\;\mathrm{negative}\;\mathrm{individual}\;\mathrm{behaviours}\right)}$$

For the latter, feeding and foraging, solitary play, object manipulation and engagement with enrichment were considered positive individual behaviours. Inactivity, self-directed and abnormal behaviours were considered negative individual behaviours (see ethogram in annex for list of positive and negative behaviours). For both indices, values varied between −1 to +1.

### 2.8. Statistical Analysis

We studied five response variables: relative frequency of affiliative behaviours, relative frequency of abnormal behaviours, relative frequency of agonistic behaviours, SCI and BCI. We carried out five separate linear mixed models (LMM) to assess the significance of treatment (T1, T2, T3, T4, T5, T6), condition (control, during treatment or post-treatment), sex (male or female), age, group (A1, B2, C3, D4), interaction group × age and time of day (evening or morning) upon the five response variables. Given that age was a continuous variable it was standardized (‘scale’ function) to facilitate the interpretation of results [30]. We included the group × age interaction because groups tended to be composed of different aged individuals. Subject identity was included as a random effect to account for having repeated measures on the same subjects [29]. Data exploration and model residuals revealed no violations to the assumptions of LMMs.

For each analysis, we created different models (‘lmer’ function) and conducted a model selection based on the corrected Akaike information criterion (AICc) (‘aictab’ function), to find the model in which the predictors offered the most parsimonious combination [31]. Next, we assessed collinearity of the predictors in the best fit model (‘vif’ function) to ensure these were not correlated [32]. We then performed a likelihood ratio test [‘anova(fit1,fit2, test = “Chisq”)’ function] to compare the best fit model to a null model, which only included the random effect, to ensure the former was a significant improvement [33]. We used the function ‘r.squaredGLMM’ to determine how much variation in the data was explained by the best fit model. We explored the model’s ANOVA table to obtain the significance of the predictors and carried out post hoc tests (Bonferroni with ‘emmeans’ and ‘emtrends’ functions) to further explore significant predictors.

Descriptive statistics (mean, median, standard deviation, maximum, minimum) for each of the response variables, grouped by condition, were also calculated.

All statistics were carried out in R [34] using the AICcmodavg [35], car [36], emmeans [37], lme4 [38] and MuMIn [39] packages. Graphs were created with ggplot2 [40].

## 3. Results

### 3.1. Summary

After carrying out approximately 225 h of observation and data collection, we were able to construct a data table with 594 entries. Table 3 includes values for the mean, median, standard deviation, maximum and minimum for each response variable. Table 4 includes the LMM results for each response variable.

### 3.2. Affiliative Behaviours

Overall, the relative frequency of affiliative behaviours ranged from 0 to 0.335, with an average of 0.055 ± 0.070 (mean ± sd) during the control condition; 0 to 0.634, with an average of 0.083 ± 0.112 during treatment; and 0 to 0.404, with an average of 0.064 ± 0.084 during post-treatment (Table 3).

The best fit model (AIC = −1334.05) (Table 5) explained 30.7% of the variance (accounting for the random effect) and included the effects of (1) condition, (2) treatment, (3) time of day, (4) sex, (5) group × age and (6) group. The variable age was not included due to issues with collinearity. The best fit model compared to the null model it was significantly better at predicting the relative frequency of affiliative behaviours (χ2 = 81.905, df = 16, *p* < 0.001).

The model revealed that condition, time of day, group and the interaction group × age had significant effects upon the relative frequency of affiliative behaviours. There was no significant effect of treatment or sex (Table 4).

Individuals displayed significantly less affiliative behaviours during the control condition compared to the treatment condition (β = −0.027, SE = 0.008, T = −3.525, *p* = 0.001). Comparison between the treatment condition and post-treatment condition revealed a marginal trend toward significance (β = 0.018, SE = 0.008, T = 2.368, *p* = 0.055). Subjects engaged in significantly more affiliative behaviours during the morning compared to the evening (β = 0.023, SE = 0.006, T = 3.654, *p* < 0.001). Seeing as the factor ‘group’ was involved in interactions, further analysis indicated it was most appropriate to only analyse said variable in conjunction with its interacting covariate, ‘age’. As such, for the average value of age, members of Group 3 displayed significantly less affiliative behaviours compared to members of the other groups (G1 vs. G3: β = 0.404, SE = 0.086, T = 4.704, *p* < 0.001), (G2 vs. G3: β = 0.351, SE = 0.079, T = 4.427, *p* < 0.001), (G3 vs. G4: β = −0.327, SE = 0.103, T = −3.169, *p* = 0.015) (Figure 1).

Thus, the predictors treatment condition, morning time and the Groups 1, 2 and 4 predicted the chimpanzees’ highest relative frequency of affiliative behaviours.

### 3.3. Abnormal Behaviours

Overall, the relative frequency of abnormal behaviours ranged from 0 to 0.365, with an average of 0.042 ± 0.064 (mean ± sd) during control; 0 to 0.249, with an average of 0.019 ± 0.005 during treatment; and 0 to 0.410, with an average of 0.043 ± 0.070 during post-treatment (Table 3).

Our selected model (AIC = −1813.68) (Table 6) explained 33.0% of the variance (including random effect) and solely included the variable (1) condition (Table 6). When compared to the null model, the selected model was significantly better at predicting the relative frequency of abnormal behaviours (χ2 = 30.177, df = 2, *p* < 0.001).

The model revealed that condition had a significant effect upon the relative frequency of abnormal behaviours (Table 4).

Individuals displayed significantly more abnormal behaviours during the control compared to during treatment (β = 0.023, SE = 0.005, T = 4.679, *p* < 0.001), and significantly less during treatment compared to post-treatment (β = −0.025, SE = 0.005, T = −4.922, *p* < 0.001).

The model predicts that the relative frequency of abnormal behaviours (including pacing, rocking, regurgitating and coprophagy) will be lowest when individuals are in the treatment condition.

### 3.4. Agonistic Behaviours

The relative frequency of agonistic behaviours ranged from 0 to 0.099, with an average of 0.008 ± 0.015 (mean ± sd) during the control condition; 0 to 0.083, with an average of 0.007 ± 0.012 during treatment; and 0 to 0.107, with an average of 0.008 ± 0.015 during post-treatment (Table 3).

The best fit model (AIC = −3460.27) (Table 7) explained 17.2% of the variance (accounting for the random effect) and included the effects of (1) condition, (2) treatment, (3) time of day, (4) sex, (5) group × age and (6) group. Compared to the null model the best fit model was significantly better at predicting the relative frequency of agonistic behaviours (χ2 = 95.6, df = 16, *p* < 0.001).

Analysis of the model showed that treatment, time of day, sex and the interaction group × age had significant effects upon the relative frequency of agonistic behaviours. Condition had no significant effect upon abnormal behaviours (Table 4).

Subjects performed significantly more agonistic behaviours during Treatment 1 compared to Treatment 4 and Treatment 6 (T1 vs. T4: β = 0.006, SE = 0.002, T = 3.332, *p* = 0.014), (T1 vs. T6: β = 0.008, SE = 0.002, T = 4.140, *p* < 0.001), and during Treatment 2 compared to Treatment 4 and Treatment 6 (T2 vs. T4: β = 0.006, SE = 0.002, T = 3.479, *p* = 0.008), (T2 vs. T4: β = 0.008, SE = 0.002, T = 4.275, *p* < 0.001). Individuals also performed more agonistic behaviours during the morning compared to the evening (β = 0.006, SE = 0.001, T = 5.953, *p* < 0.001). Females engaged in significantly less agonistic behaviours relative to males (β = −0.005, SE = 0.001, T = −3.431, *p* = 0.005). For the average value of age, members of Group 2 displayed significantly less agonistic behaviours than members of Group 3 (G2 vs. G3: β = −0.029, SE = 0.009, T = −3.081, *p* = 0.034) (Figure 2).

Overall, the predictors treatment (4 and 6), evening time, group (2) and sex (female) predicted chimpanzees’ lowest relative frequency of agonistic behaviours.

### 3.5. Social Competence

SCI ranged from −1 to −0.311, with an average of −0.870 ± 0.140 (mean ± sd) during the control condition; −1 to 0.268, with an average of −0.815 ± 0.227 during treatment; and −1 to −0.220, with an average of −0.856 ± 0.168 during post-treatment (Table 3).

The best fit model (AIC = −485.05) (Table 8) explained 28.1% of the variance (with random effect) and was composed of the following factors: (1) condition, (2) treatment, (3) time of day, (4) sex, (5) group × age and (6) group and was significantly better at predicting the SCI compared to the null model (χ2 = 83.873, df = 16, *p* < 0.001).

The model’s ANOVA table revealed that condition, time of day, group and the interaction group × age had significant effects upon the subjects’ SCI. Treatment and sex did not present significant effects (Table 4).

Individuals had a significantly lower SCI during the control condition compared to the treatment condition (β = −0.054, SE = 0.016, T = −3.406, *p* = 0.002), and a significantly higher SCI during treatment compared to post-treatment (β = 0.040, SE = 0.016, T = 2.520, *p* = 0.036). SCI was significantly higher during the morning than the evening (β = 0.057, SE = 0.013, T = 4.422, *p* < 0.001). For the average value of age, members of Group 3 presented significantly lower levels of SCI compared to members of the other groups (G1 vs. G3: β = 0.785, SE = 0.162, T = 4.835, *p* < 0.001), (G2 vs. G3: β = 0.677, SE = 0.150, T = 4.506, *p* < 0.001), (G3 vs. G4: β = −0.565, SE = 0.197, T = −2.866, *p* = 0.032) (Figure 3).

Thus, the model predicts that individuals will present a higher level of social competence during the treatment condition, in the morning and when belonging to Group 1, 2 or 4.

### 3.6. Behavioural Competence

BCI ranged from −1 to 0.950, with an average of −0.241 ± 0.561 (mean ± sd) during the control condition; −0.936 to 1, with an average of 0.272 ± 0.495 during treatment; and −1 to 0.924, with an average of −0.245 ± 0.554 during post-treatment (Table 3).

The best fit model (AIC = 443.47) (Table 9) explained 67.4% of the variance (with random effect) and was composed of: (1) condition, (2) treatment, (3) time of day, (4) sex, (5) group × age and (6) group and was significantly better at predicting the BCI compared to the null model (χ2 = 417.32, df = 16, *p* < 0.001).

Condition, treatment, time of day, sex, group and the interaction group × age all had significant effects upon the subjects’ BCI (Table 4).

Individuals had a significantly lower BCI during the control compared to the treatment condition (β = −0.507, SE = 0.034, T = −14.894, *p* < 0.001), and a significantly higher BCI during treatment compared to post-treatment (β = 0.510, SE = 0.034, T = 14.991, *p* < 0.001). During Treatment 1 individuals displayed a significantly lower BCI compared to all other treatments (T1 vs. T2: β = −0.145, SE = 0.046, T = −3.133, *p* = 0.027), (T1 vs. T3: β = −0.184, SE = 0.047, T = −3.949, *p* = 0.001), (T1 vs. T4: β = −0.357, SE = 0.047, T = −7.561, *p* < 0.001), (T1 vs. T5: β = −0.201, SE = 0.051, T = −3.937, *p* = 0.001), (T1 vs. T6: β = −0.281, SE = 0.051, T = −5.549, *p* < 0.001). Moreover, during Treatment 4 individuals displayed a significantly higher BCI than during Treatments 2, 3 and 5 (T2 vs. T4: β = −0.213, SE = 0.047, T = −4.503, *p* < 0.001), (T3 vs. T4: β = −0.173, SE = 0.047, T = −3.665, *p* = 0.004), (T4 vs. T5: β = 0.157, SE = 0.050, T = 3.118, *p* = 0.029). BCI was significantly higher during the morning compared to the evening (β = 0.316, SE = 0.028, T = 11.331, *p* < 0.001), and females displayed significantly lower levels of BCI than males (β = −0.189, SE = 0.066, T = −2.848, *p* = 0.008). For the average value of age, members of Group 3 presented significantly lower levels of BCI compared to the other groups (G1 vs. G3: β = 2.957, SE = 0.471, T = 6.276, *p* < 0.001), (G2 vs. G3: β = 2.932, SE = 0.434, T = 6.751, *p* < 0.001), (G3 vs. G4: β = −3.466, SE = 0.555, T = −6.245, *p* < 0.001) (Figure 4).

Thus, the model predicts that individuals will present a higher level of behavioural competence during the treatment condition, especially during treatment 4, in the morning, when subjects are male and belong to Group 1, 2 or 4.

## 4. Discussion

The results of this study revealed that the 20 chimpanzees observed at TCS, performed more affiliative and abnormal behaviours relative to agonistic behaviours. From a social perspective this indicates that the subjects spend relatively more time engaging in positive interactions than negative ones: a trend that intensified with the presence of nesting materials, given that affiliation further increased and agonism further decreased. Although abnormal behaviours were common throughout the study, the presence of bedding led to a decrease in abnormality. Both welfare indices (SCI and BCI) were low. Negative values of SCI suggest that individuals are performing more individual behaviours than social behaviours (possibly due to groups being formed of similar aged individuals, who do not seek their peers for all social interaction). Negative values of BCI suggest individuals are performing more negative individual behaviours (e.g., self-directed, abnormal) than positive ones (e.g., feeding, object manipulation). Nevertheless, when presented with bedding, the subjects’ SCI and BCI increased, meaning the presence of this enrichment offered opportunities to dedicate less time to performing individual and negative behaviours.

Our study suggests that welfare increases during the treatment condition, relative to both control and post-treatment, thus supporting our first prediction that welfare improves in the presence of nesting materials. This finding was evidenced by the relative increase of affiliative behaviours, decrease of abnormal behaviours and increased SCI and BCI during the treatment condition. Although the relative frequency of agonistic behaviours did not differ substantially between conditions, seeing as the presence of bedding did not result in increased agonistic behaviours, we can conclude that welfare improved on four out of five measures. Other studies have also shown that primate welfare increases in the presence of bedding [14], with both Baker (1997) [17] and Brent (1992) [16] reporting a decline in abnormal behaviours in captive chimpanzees.

Our second prediction that welfare would be most positively affected by Treatment 1 (leaves and branches) was not supported. Only the relative frequency of agonistic behaviours and BCI changed under different treatments, with all other response variables remaining unchanged. The relative frequency of agonistic behaviours was lowest during Treatment 4 (shredded newspaper) and Treatment 6 (cotton sheets plus shredded newspaper), and BCI was highest during Treatment 4, suggesting that chimpanzees’ welfare is higher when presented with shredded newspaper. Although leaves and branches are used by wild chimpanzees during nest building [22,23] we deduce that captive individuals engage in many more positive behaviours than simply building a nest when provisioned with shredded newspaper. Common positive behaviours observed with this material were solitary and social play, chewing, mouth tearing and hand manipulation (we also noted that even during other treatments, nest building was only one of several behaviours exhibited). A study by Videan and colleagues (2005) [41] reported a similar finding, whereby captive chimpanzees spent most time engaged with enrichment items that were destructible, including newspaper, as these were believed to offer individuals more controllability over their environment. Moreover, newspaper is a form of synthetic material that cannot be found in the outdoor enclosures, whereas branches, leaves, twigs and small patches of grass can indeed be found. Therefore, it is plausible that the chimpanzees also showed improved welfare with newspaper due to its novelty and uniqueness.

Our third prediction that a combination of nesting materials would yield a more positive impact upon welfare was also not supported. There was no difference between the measures of welfare during Treatment 5 (leaves and branches plus long grass) compared to Treatments 1 and 2 (long grass), or between Treatment 6 compared to Treatments 3 (cotton sheets) and 4. Contrary to the recent recommendation of providing captive great apes with multiple nesting materials where at least one is known to be preferred [21], we found no substantial difference in the chimpanzees’ welfare when presented exclusively with shredded newspaper and when the newspaper was presented with an additional material. Our finding suggests that the welfare of captive chimpanzees is primarily dependent on the presence of a preferred material, regardless of whether it is presented solely or in conjunction with another.

Prediction 4 that welfare would be higher in the evening compared to the morning was not supported, seeing as three out of the five welfare parameters (affiliative behaviours, SCI and BCI) showed relatively higher values during the morning. Although individuals made first contact with the nesting materials upon entry indoors at the end of the day, we believe their engagement level was lessened due to an increase in time spent resting. Therefore, bedding had a more positive impact upon welfare during the morning because individuals had more energy to engage with said enrichment after resting through the night. Although the relative frequency of abnormal behaviours remained unchanged, the relative frequency of agonistic behaviours increased during the morning, but was not affected by condition. This suggests that individuals’ always display relatively higher levels of agonism during the morning and, as such, the presence of the materials does not result in a detrimental increase in agonistic behaviours. Previous studies have also noted that captive chimpanzees display relatively more agonistic behaviours in the morning [28,29] and, in our study, this is likely linked to individuals’ impatience to be released outside.

Prediction 5, whereby we expected females to show higher levels of welfare due to spending more time building nests, was not supported. Although the presence of bedding encouraged females to engage in less agonistic behaviours compared to males, the latter engaged in more positive individual behaviours (relative to social behaviours i.e., BCI) than females. This suggests that males spent relatively more time manipulating, tearing and exploring the bedding materials than their female counterparts. Previous investigators have also found agonistic behaviour to be low in captive female chimpanzees [29]. Nevertheless, future studies could further investigate sex differences regarding engagement with bedding enrichment, to ascertain whether these differences are important and should be considered when implementing enrichment programmes.

We did not explore prediction 6 that younger individuals would display higher levels of welfare compared to older individuals, because the variable age was not included in any of the final models. Nevertheless, prediction 7 that different groups would display different levels of welfare was confirmed. Despite the small age range between the study subjects (simply due to those being the subjects Tacugama gave the researchers access to), prediction 8 that groups composed of younger individuals would display higher levels of welfare compared to groups composed of older individuals (i.e., group × age interaction) was not supported. In fact, the group composed of the youngest individuals (Group 3) presented poorer welfare on three out of the five measures: inferior relative frequency of affiliative behaviours, lower BCI and lower SCI compared to all other groups. This finding is possibly related to infants having suffered a more recent loss of their mother (seeing as most individuals become orphans and arrive at TCS as babies), and not having an older group member to act as replacement to offer reassurance, support and observational learning opportunities, e.g., tool use [42]. An investigation by Reddy and Mitani (2019) [43], showed that older chimpanzees often adopt younger siblings following the death of their mother, providing them with the necessary care and skills to survive. Furthermore, Boesch et al. (2010) [44] found that unrelated group members also adopt younger orphaned chimpanzees. These studies indicate that older chimpanzees do in fact care for younger individuals, who require the comfort and knowledge of the more experienced to learn all the characteristic skills involved in becoming a chimpanzee. In this way, the infants of Group 3 likely displayed relatively lower welfare levels compared to the juveniles and young adults because they lack more mature companions who can reassure them, as well as teach them how to take full advantage of the enrichment provided. Future investigations would benefit from including older chimpanzees, to not only assess differences between infants and juveniles/young adults but also between these and mature adults.

It would also be valuable for future studies to conduct a similar type of investigation and assess whether the provision of one or multiple nesting types have differing impacts upon chimpanzee welfare or not. Although there were a few difficulties in carrying out the proposed methods, considering that our experimental design can be easily repeated, it would also be beneficial to repeat this investigation and ensure data collection was carried out over consecutive days (without any missing days between conditions), as well as obtaining the full number of scans per condition, in order to fully confirm our findings.

Seeing as this type of study generates a wide range of results, we believe the collection of similar data over large time frames would also offer the opportunity to investigate the evolution of captive individuals’ rates of affiliation, abnormality and agonism, as well as their social and behavioural competence. Data on these variables would enable captive facilities to scientifically assess whether they are successfully caring and attending to all of their residents’ needs, and therefore identify behaviours that are lacking and need encouraging, behaviours that are negative and need reducing, as well as behaviours that are positive and should be maintained.

## 5. Conclusions

After a lack of recent studies investigating the impact of different types of nesting materials upon the welfare of chimpanzees, our research provides an updated confirmation of the positive effects bedding materials produce upon rehabilitation individuals, through the analysis of a variety of welfare measures. Although all materials improved welfare, the most favourable was shredded newspaper, regardless of whether it was presented separately or in conjunction with another material. This finding shows that synthetic enrichments can indeed have a more positive effect upon chimpanzee behaviour than natural ones. In this case, shredded newspaper was likely the most favourable due to its novelty and the fact it offers rehabilitated chimpanzees more control over their surroundings, and thus enabling individuals to engage in a multitude of behaviours beyond nest building. The materials had a more positive impact upon welfare during the morning, compared to the evening, possibly because individuals are more active and willing to engage with the materials post-rest. Both sexes experienced heightened welfare with bedding, with females displaying relatively less agonistic behaviours and males engaging in more exploratory actions towards the enrichment. Although welfare increased for groups of all ages, those composed of younger individuals did not achieve the welfare levels displayed by groups with older chimpanzees, most likely due to a shorter period of rehabilitation and a lack of role models from which to receive support and life lessons.

Based on our findings, we suggest that captive chimpanzees (of both sexes whose age falls within the range sampled) be provisioned with nesting materials when in indoor enclosures (at all times of day), with materials that offer destructibility and changeability (e.g., can be teared or rearranged into diverse patterns), to ensure high levels of welfare are continuously achieved.

## Figures and Tables

**Figure 1 animals-11-01835-f001:**
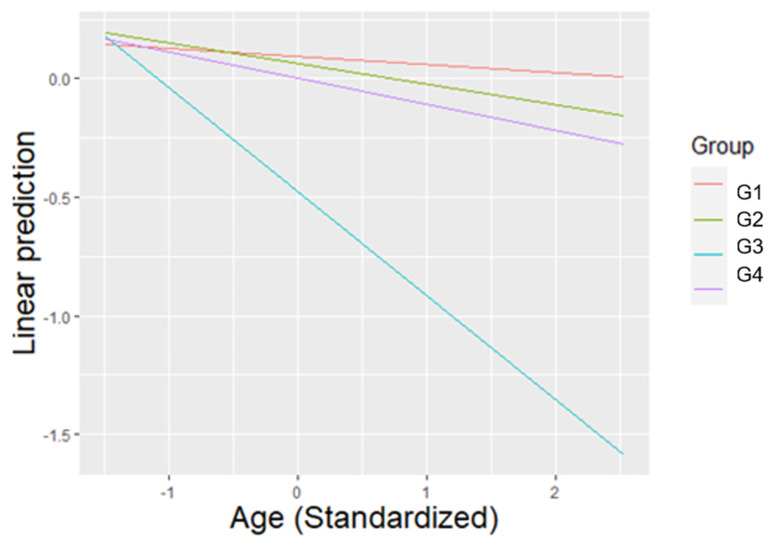
The linear prediction of the relative frequency of affiliative behaviours according to the average value of age (previously standardized), for each group. For the average value of age, Group 3 displayed significantly less affiliative behaviours than all other groups.

**Figure 2 animals-11-01835-f002:**
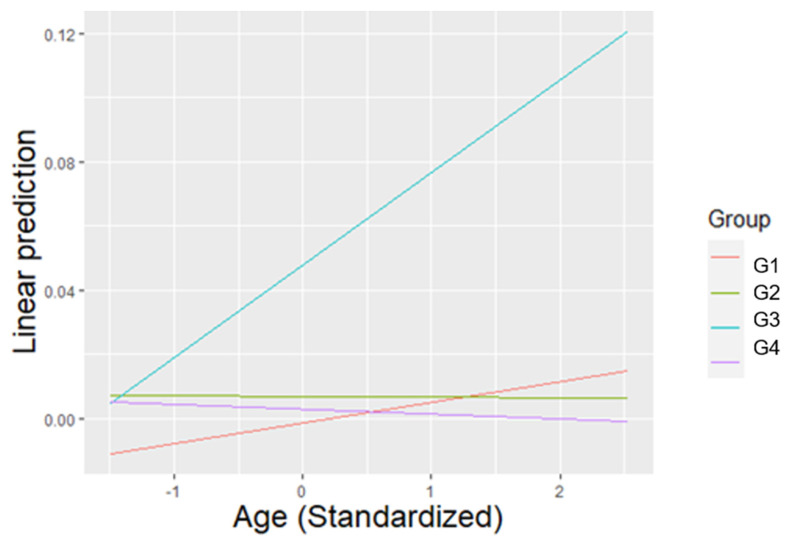
The linear prediction of the relative frequency of agonistic behaviours according to the average value of age (previously standardized), for each group. For the average value of age, Group 3 displayed significantly more agonistic behaviours than group 2.

**Figure 3 animals-11-01835-f003:**
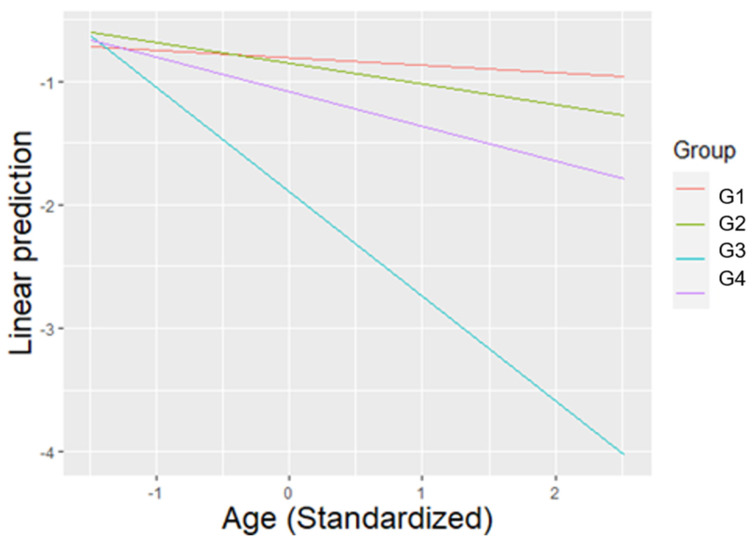
The linear prediction for SCI according to the average value of age (previously standardized), for each group. For the average value of age, Group 3 presented significantly lower SCI than all other groups.

**Figure 4 animals-11-01835-f004:**
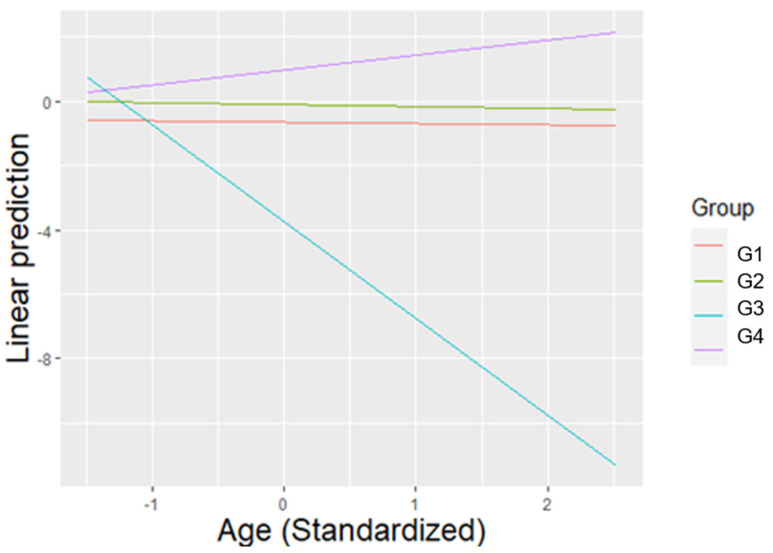
The linear prediction of BCI according to the average value of age (previously standardized), for each group. For the average value of age, Group 3 presented significantly lower behavioural competence compared to all other groups.

**Table 1 animals-11-01835-t001:** Summary of all study subjects and relevant characteristics.

Subject	Age	Sex	Group
Mac	13	Male	G1
Nico	15	Male	G1
Monko	8	Female	G2
Hashi	8	Female	G2
Winnie	8	Female	G2
Lola	9	Female	G2
Kortu	8	Female	G2
Perry	9	Male	G2
Mortes	9	Male	G2
Tetina	8	Male	G2
Molly	9	Female	G2
Mus	9	Female	G2
Michael	7	Male	G2
Morlai	8	Male	G2
Jean	5	Female	G3
Joko	4	Male	G3
Bousy	4	Male	G3
Sophie	5	Female	G4
Matilda	5	Female	G4
Woron	4	Male	G4

**Table 2 animals-11-01835-t002:** Types of bedding materials used during the study.

Treatment	Type of Bedding
1	Leaves and branches
2	Long grass
3	Cotton sheets
4	Shredded newspaper
5	Leaves and branches + Long grass
6	Cotton sheets + Shredded newspaper

**Table 3 animals-11-01835-t003:** Descriptive statistics for each response variable, grouped by condition, regarding the behaviour of the chimpanzees at Tacugama Chimpanzee Sanctuary (data collected from November 2019 until March 2020).

Response Variable	Condition	Mean	Median	St. Deviation	Maximum	Minimum
Rel. Freq. of Affiliative	Control	0.055	0.025	0.070	0.335	0
	During Treatment	0.083	0.030	0.112	0.634	0
	Post-Treatment	0.064	0.030	0.084	0.404	0
Rel. Freq. of Abnormal	Control	0.042	0.016	0.064	0.365	0
	During Treatment	0.019	0.005	0.038	0.249	0
	Post-Treatment	0.043	0.015	0.070	0.410	0
Rel. Freq. of Agonistic	Control	0.008	0	0.015	0.099	0
	During Treatment	0.007	0	0.012	0.083	0
	Post-Treatment	0.008	0	0.015	0.107	0
SCI	Control	−0.870	−0.923	0.140	−0.311	−1
	During Treatment	−0.815	−0.905	0.227	0.268	−1
	Post-Treatment	−0.856	−0.927	0.168	−0.220	−1
BCI	Control	−0.241	−0.342	0.561	0.950	−1
	During Treatment	0.272	0.359	0.495	1	−0.936
	Post-Treatment	−0.245	−0.345	0.554	0.924	−1

**Table 4 animals-11-01835-t004:** Overall results of the linear mixed models for each response variable regarding the behaviour of the chimpanzees at Tacugama Chimpanzee Sanctuary (data collected from November 2019 until March 2020).

Response Variable	Independent Variable	*F*	df	*p*	*V* (%)
Rel. Freq. of Affiliative	Condition	6.566	2	0.002 **	30.7
	Treatment	1.807	5	0.1095	-
	Time of Day	13.559	1	0.000 ***	-
	Sex	0.808	1	0.379	-
	Group	15.202	3	0.000 ***	-
	Group × Age	17.762	4	0.000 ***	-
Rel. Freq. of Abnormal	Condition	15.498	2	0.000 ***	33.0
Rel. Freq. of Agonistic	Condition	0.404	2	0.668	17.2
	Treatment	6.219	5	0.000 ***	-
	Time of Day	36.483	1	0.000 ***	-
	Sex	12.194	1	0.001 ***	-
	Group	4.069	3	0.007 **	-
	Group × Age	3.233	4	0.012 *	-
Social Competence Index	Condition	6.353	2	0.002 **	28.1
	Treatment	0.894	5	0.485	-
	Time of Day	19.859	1	0.000 ***	-
	Sex	0.353	1	0.559	-
	Group	15.419	3	0.000 ***	-
	Group × Age	18.544	4	0.000 ***	-
Behavioural Competence Index	Condition	151.544	2	0.000 ***	67.4
	Treatment	12.909	5	0.000 ***	-
	Time of Day	130.421	1	0.000 ***	-
	Sex	13.996	1	0.000 ***	-
	Group	28.550	3	0.000 ***	-
	Group × Age	23.645	4	0.000 ***	-

*F* corresponds to the F-statistic, df to the degrees of freedom, *p* to the *p*-Value where * is a significance of <0.05, ** is a significance of <0.01 and *** is a significance of <0.001, and *V* specifies the amount of variation explained by the model.

**Table 5 animals-11-01835-t005:** Model structures and AIC values obtained during model selection for affiliative behaviours.

Model	Model Structure	AICc
Full Model	Condition, treatment, time of day, sex, group, group × age, random effect for subject identity	−1333.05
Model 1	Condition, treatment, time of day, sex, group × age, random effect for subject identity	−1310.08
Model 2	Condition, treatment, time of day, sex, random effect for subject identity	−1298.67
Model 3	Condition, treatment, time of day, random effect for subject identity	−1300.37
Model 4	Condition, treatment, random effect for subject identity	−1289.37
Model 5	Condition, random effect for subject identity	−1293.12
Null Model	random effect for subject identity	−1284.43

AICc is the corrected Akaike information criterion used to evaluate which is the best fit model. Lower AICc values indicate a more parsimonious model.

**Table 6 animals-11-01835-t006:** Model structures and AIC values obtained during model selection for abnormal behaviours.

Model	Model Structure	AICc
Full Model	Condition, treatment, time of day, sex, group, group × age, random effect for subject identity	−1810.52
Model 1	Condition, treatment, time of day, sex, group × age, random effect for subject identity	−1807.56
Model 2	Condition, treatment, time of day, sex, random effect for subject identity	−1805.61
Model 3	Condition, treatment, time of day, random effect for subject identity	−1807.57
Model 4	Condition, treatment, random effect for subject identity	−1807.61
Model 5	Condition, random effect for subject identity	−1813.68
Null Model	random effect for subject identity	−1787.57

AICc is the corrected Akaike information criterion used to evaluate which is the best fit model. Lower AICc values indicate a more parsimonious model.

**Table 7 animals-11-01835-t007:** Model structures and AIC values obtained during model selection for agonistic behaviours.

Model	Model Structure	AICc
Full Model	Condition, treatment, time of day, sex, group, group × age, random effect for subject identity	−3460.27
Model 1	Condition, treatment, time of day, sex, group × age, random effect for subject identity	−3456.21
Model 2	Condition, treatment, time of day, sex, random effect for subject identity	−3451.75
Model 3	Condition, treatment, time of day, random effect for subject identity	−3445.16
Model 4	Condition, treatment, random effect for subject identity	−3412.33
Model 5	Condition, random effect for subject identity	−3394.79
Null Model	random effect for subject identity	−3397.95

AICc is the corrected Akaike information criterion used to evaluate which is the best fit model. Lower AICc values indicate a more parsimonious model.

**Table 8 animals-11-01835-t008:** Model structures and AIC values obtained during model selection for social competence.

Model	Model Structure	AICc
Full Model	Condition, treatment, time of day, sex, group, group × age, random effect for subject identity	−485.05
Model 1	Condition, treatment, time of day, sex, Group × age, random effect for subject identity	−461.36
Model 2	Condition, treatment, time of day, sex, random effect for subject identity	−451.27
Model 3	Condition, treatment, time of day, random effect for subject identity	−452.43
Model 4	Condition, treatment, random effect for subject identity	−435.40
Model 5	Condition, random effect for subject identity	−442.59
Null Model	random effect for subject identity	−434.46

AICc is the corrected Akaike information criterion used to evaluate which is the best fit model. Lower AICc values indicate a more parsimonious model.

**Table 9 animals-11-01835-t009:** Model structures and AIC values obtained during model selection for behavioural competence.

Model	Model Structure	AICc
Full Model	Condition, treatment, time of day, sex, group, group × age, random effect for subject identity	443.47
Model 1	Condition, treatment, time of day, sex, group × age, random effect for subject identity	471.41
Model 2	Condition, treatment, time of day, sex, random effect for subject identity	490.05
Model 3	Condition, treatment, time of day, random effect for subject identity	489.05
Model 4	Condition, treatment, random effect for subject identity	603.74
Model 5	Condition, random effect for subject identity	638.75
Null Model	random effect for subject identity	827.51

AICc is the corrected Akaike information criterion used to evaluate which is the best fit model. Lower AICc values indicate a more parsimonious model.

## Data Availability

Data supporting this study’s findings are available by fair request from the corresponding author.

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
