# Peer review of "An Investigation into the Influence of Different Types of Nesting Materials upon the Welfare of Captive Chimpanzees (Pan troglodytes)"

_animals, 2021, doi:10.3390/ani11061835_

Round 1
Reviewer 1 Report
Thank you for the opportunity to read and to comment on the article. This project is interesting and well designed, with application to individual welfare and to broader efforts to nurture the natural behavioral repertoire of captive chimpanzees. The research design enables measurement of the durability of the intervention, and it could highlight individual chimpanzees who respond differently to the enrichment, which is a useful tool for others caring for captive chimpanzees. The method is also replicable in other settings and could be collected via video—important in these times of heightened concerns about disease transmission. The paper is clearly and well written and argued. My suggestions are minor.
L21 italicize Linnaean name, change writing to “…in part because they can act…” (because as you note later in the paper, welfare is also important for the individual)
L41 “plenty” is arguable—I suggest deleting the word.
L48 “wholesome” is arguable—I suggest deleting the word.
L54 I suggest you delete “properly.”
L59 delete “in particular” or rewrite to read “The provision of bedding is one way…”and throughout paragraph check Linnaean names—I believe at first reference the Linn name should be provided for species (e.g., rhesus macaques)
L134 “means”: elaborate on this: staff, funds, cleaning (probably all of these)
L134 and L150-158 Please provide more information on the bedding: is the bedding enrichment fresh and new at each introduction? Is any other enrichment provided throughout the day? Does the bedding remain in the enclosure for some period of time, or was it cleared out after 24 hours (or prior to each data collection session)?
L137 It seems that you might have data to compare the treatment days to days when no bedding was provided at all. This comparison might show a stronger positive welfare impact of bedding.
In your figures 1-4, I became confused about which chimpanzees are in the groups depicted (or what features characterize groups)—would the same individuals be in group 1 in Figure 1 and in Figure 2, or are the data being reconfigured? And these groups are not the housing groups? Perhaps some clarification here will be helpful to other readers, too.
It might help to provide more information about how the bedding enrichment is distributed. Are some materials easier for a chimpanzee to monopolize? Or are the chimpanzees shifted over and the bedding is distributed after cleaning? It seems from the article that each chimpanzee was able to access the bedding, but I am curious whether agonism might increase when the bedding is introduced and how even distribution of enrichment is accomplished.
L428 and forward: You might want to discuss here that the chimpanzees in the study could not/did not receive fresh (or any?) bedding on a daily basis, so the response might be different for chimpanzees that receive bedding daily (or even more often). I imagine bedding is more enriching to the lives of those who less frequently receive it. If this study compared bedding days to no-bedding days, a positive impact on welfare might have been detected (and see my comment above—this would be interesting to add to the model if there is comparable data on no-bedding days).
Author Response
Comments to Reviewer 1:
We are very thankful for your thoughtful suggestions and understand and agree with the points you make. Thanks for taking the time to go through the paper, it is my MSc thesis – the presentation went well and we were encourage to try and publish. We are working very hard to achieve that goal! Please see the comments below to each of your points:
L21 – changes made.
L41 – changes made.
L48 – changes made.
L54 – changes made.
L59 – changes made.
L134 – elaborated on “means”
L134 – provided further explanation on the presentation of the bedding materials
L137 - instead of doing the two-way comparison of days with bedding and days without, we focused on a comparison that included the before and after conditions as well, as this was what we were really interested in. We wanted to assess whether the effects of enrichment could have any lasting positive effects on welfare even after it was cleared from the enclosure. This point was not discussed as it did not happen.
Figures 1 to 4 – Apologies for the confusion. We have changed the figure legends to match the groups, I hope this is clear now. The groups tested are the housing groups presented in Table 1.
Distribution of enrichment – We have included in the materials & methods a quick sentence that bedding was added to the enclosure before the chimps entered. It was scattered around so all chimps would be able to run inside and get a piece for themselves. We didn’t actually have issues with complete monopoly of the bedding, although some chimps would end up with more and others with less, but all had access to bedding. Perhaps because the juveniles don’t have a clear dominant and the infants are still young, the monopoly of the enrichment was not a major concern. For the two young adult males, because it is only two in a group who sleep separately it was easy to ensure they both had access to the materials as well.
L428 – Yes, bedding days vs non bedding days would definitely be interesting to include in the model. However, as mentioned above I think the split into three actually offers more insights and interesting results.
Reviewer 2 Report
This paper studies the importance of different types of enrichment on animal welfare, in this case of chimpanzees. I do not consider the provided items as bedding, but more enrichment (mainly to play and perhaps form nests). I would therefore propose to adjust the title (or the authors must specify in the introduction why they see this as bedding). Although not new or surprising, this topic can be of interest to rescue centers or other organisations that house captive chimpanzees. Although the authors refer to older and younger animals, most animals are younger than 10 years and 2 are 15 and 13. This is not a group representing different ages, they are all quite young.The manuscript is quite long for the data and message provided. Shortening the text and focus on the most important findings would benefit the paper.
Specific remarks
Abstract: line 32 The authors state that in groups composed of older individuals (vs younger) welfare was highest in the morning. I cannot read this from the results. Also the age difference is really limited (and there are mainly young animals) and the groups are small (see remarks below as well).
Material and Methods:
General remark: I miss an outline of the general set-up of TCS, i.e. indoor facilities/outdoor facilities, set-up of the different groups at TCS (including over-all age-range). Especially the enrichment they have in outdoor facilities should be provided as well to exclude potential differences between groups.
Table 1: The authors selected 20 animals within a limited age range (all relatively young animals). It is not clear what the criteria for the selection of these animals were. TCR houses 8 groups with a totla of 99 animals (line 143). Are the animals mentioned in Table 1 part of a larger group? If so, why are these animals selected, what was their status in the group and what was the composition of that group? Are there no animals older than 15?Did the authors randmomize for the time that they were already in the rescue center to avoid differences in animals that were recently rescued and animals that were already more adapted to the more favorable situations at TCS. Different situations can have enourmous effects on behaviour of the animals.
Table 1: The authors use G1 to G4, in the Figures they use A1, B2 etc. Is this the same. If so, please use the same identification, if this is not the same, please specify what is A to D.
Methods line 204-208. The authors check welfare of the animals based upon behaviour, including food manipulation, foraging and engagement with enrichment items. These different effects are included in the SCI and BCI. Although this is certainly a good way, it would be of interest to see how the different groups handle the different items (leaves, grass, cotton sheets, newspaper) and if there are differences between the groups in the use of these items.
Table 2: Did the authors consider to mix natural enrichment items (which the animals are probably familiar with) with non-natural enrichment (newspaper of cotton sheets) to see if that makes a difference? (see also discussion lines 433-436)
Results (3.3): Could the authors specify "abnormal behaviour"
Discussion: 415-421: The effect of newspaper on welfare was higher that that of grass or leaves. This might be due to the relative novelty of the newspapers/cotton sheets. Do the animals have leaves and twigs and/or grass in their outside enclosure? Do the authors have information on the use and on welfare when the animals have received this material for longer time periods and the novelty wears off?
Do the authors have data from older chimpanzees as well?
Author Response
Comments to Reviewer Two
Thank you for your comments. We gladly take them on board to improve the paper. This is my MSc thesis and as I have never published before, we are working hard to see if that is an option. We have improved on all comments (incluiding the change from “bedding” to “nesting materials”), please see point-by-point explanations below:
Abstract line: The intention was not to say that older individuals are the ones with higher welfare in the morning, but to list all variables included in our statistical model and quickly show the result for each. I.e. regarding time of day, welfare was higher in the morning; regarding age, welfare was higher for individuals of older ages. We have tried to re-phrase so this is the message that comes through.
Outline of TCS: Added into materials & methods section.
Table 1 Age: We did not have much choice re selection of individuals to analyse. Although yes there were 99 chimps, some were in further enclosures where it was not safe to be alone or in early/late hours. I (Naomi) didn’t have anyone to accompany me during data collection and got chased by a wild chimp that hangs around the sanctuary on a few occasions. After data was collected for the juveniles the idea was to go to larger groups with more age range e.g. 2 – 30 years (yes, this would have been great) but due to safety I had no choice but to move to the baby section as it is closer to the centre of the sanctuary with more people around. The animals analysed made up the entirety of each group. G1 only had 2 individuals due to failure to integrate them with others; G2 is the juvenile group; G3 at the time was a subgroup of the nursery section with only 3 babies that had been integrated together; G4 was another subgroup of the nursery section. We did not analyse time since rescue in the model, we appreciate this can have an impact hence it was a discussion point in the discussion.
Table 1 Groups: Apologies, yes same groups. Figure legend corrected.
Handling of materials: Agreed, a very interesting point. We did not analyse exactly which behaviours were performed more with which bedding type. However, we did mention that newspaper likely enabled the chimps to engage in a larger variety of behaviours.
Mixing materials: Yes we considered mixing 1natural with 1 synthetic – due to time we did not. If we ever repeat the study this is something we would do!
Specify abnormal behaviours: Done.
Novelty: Point added in discussion. We do not have further long-term data on novelty wearing off as it was the first enrichment study with nesting materials conducted at Tacugama.
We don’t have data for older chimpanzees. As mentioned above we are aware this would improve our results.
Regarding the overall length of the manuscript, we tried to reduce the size of the intro by cutting out some references. However, we are struggling to decide where more words should be removed. If you still feel strongly that the paper is too long we would be happy to take on board suggestions of where exactly to reduce word count.
Round 2
Reviewer 1 Report
Thank you for the opportunity to review your manuscript, I enjoyed seeing your results!
Reviewer 2 Report
The manuscript has been improved. I can accept that in a santuary it is not always possible to perform optimally designed studies. However, this should be well-discussed and taken on board beforehand when designing future studies in such institutes to prevent that data cannot be interpreted optimally.